# Gender equality related to gender differences in life expectancy across the globe gender equality and life expectancy

**Ana-Catarina Pinho-Gomes**[1,2]*, **Sanne A. E. Peters**[1,3,4], **Mark Woodward**[1,3]

**1** The George Institute for Global Health, Imperial College London, London, United Kingdom, **2** School of Population Health & Environmental Sciences, Faculty of Life Sciences & Medicine, King's College London, London, United Kingdom, **3** The George Institute for Global Health, University of New South Wales, Sydney, New South Wales, Australia, **4** Julius Center for Health Sciences and Primary Care, University Medical Center Utrecht, Utrecht University, Utrecht, The Netherlands

* a.pinho-gomes@imperial.ac.uk

**Data Availability Statement:** All data required to replicate this study are available within the manuscript and supplementary data or shared in a

## Abstract

Life expectancy (LE) depends on the wider determinants of health, many of which have gendered effects worldwide. Therefore, this study aimed to investigate whether gender equality was associated with LE for women and men and the gender gap in LE across the globe. Gender equality in 156 countries was estimated using a modified global gender gap index (mGGGI), based on the index developed by the World Economic Forum between 2010 and 2021. Linear regression was used to investigate the association between the mGGGI and its economic, political, and education subindices and the gender gap in LE and women and men's LE. Overall, the mGGGI increased from 58% in 2010 to 62% in 2021. Globally, changes in the mGGGI and its economic and political subindexes were not associated with changes in the gender gap in LE or with LE for women and men between 2010 and 2020. Improvements in gender equality in education were associated with a longer LE for women and men and widening of the gender gap in LE. In 2021, each 10% increase in the mGGGI was associated with a 4.3-month increase in women's LE and a 3.5-month increase in men's LE, and thus with an 8-month wider gender gap. However, the direction and magnitude of these associations varied between regions. Each 10% increase in the mGGGI was associated with a 6-month narrower gender gap in high-income countries, and a 13- and 16-month wider gender gap in South and Southeast Asia and Oceania, and in Sub-Saharan Africa, respectively. Globally, greater gender equality is associated with longer LE for both women and men and a widening of the gender gap in LE. The variation in this association across world regions suggests that gender equality may change as countries progress towards socioeconomic development and gender equality.

## Introduction

Notwithstanding substantial progress made over the past centuries, gender equality in all spheres of private and societal life remains far off and fulfilment of women's rights is lagging worldwide [1]. Gender equality implies that the interests, needs and priorities of both women

public GitHub repository (https://github.com/Ana-Catarina/global-gender-gap.git).

**Funding:** The authors received no specific funding for this work.

**Competing interests:** I have read the journal's policy and the authors of this manuscript have the following competing interests: MW is a consultant for Amgen, Kyowa Kirin and Freeline. There are no patents, products in development or marketed products associated with this research to declare. This does not alter our adherence to PLOS ONE policies on sharing data and materials.

and men are taken into consideration, thereby recognising the diversity of different groups of women and men [2]. It does not mean that women and men will become the same but that women's and men's rights, responsibilities and opportunities will not depend on whether they are born female or male. According to the United Nations, equality between women and men is seen both as a human rights issue and as a precondition for, and indicator of, sustainable people-centred development [3].

Life expectancy (LE) at birth is a well-established indicator of the overall health of a population [4, 5]. LE reflects the cumulative effect of biological, behavioural and environmental determinants of health, including working and living conditions, exposure to pollution, access to health care, education, income, and social support [6]. Many of these wider determinants have a gendered impact, which underpins the longstanding gender gap in LE [7]. For instance, women accounted for only 38 percent of worldwide human capital wealth in 2018 [8], and such an inequality may contribute to adverse health outcomes [9]. Climate change is also likely to have a larger impact on women than men and also carry more adverse health consequences for women than men [10]. Crucially, these factors are influenced by gender policies across multiple governmental sectors, such as economy, education, and social welfare [11].

Although women have historically outlived men worldwide [12], the gender gap in LE increased from 4.5 years in 1950 to 5.4 years in 2012, then decreased to 5.1 years by 2019 [12]. Within these global comparisons, there are marked variations between world regions. For instance, the gender gap was 8.6 years in central Europe, eastern Europe and central Asia and 3.2 years in North Africa and Middle East in 2019 [12]. Biologic factors, such as sex hormones, may partially account for the longer LE of women in comparison to men [13]. However, if the gender gap was explained exclusively by biological factors, the gap would remain relatively constant instead of varying substantially over time and place [14]. A detailed demographic analysis showed that prior to 1950 excess mortality among baby boys was the main driver of the gender gap in LE, whilst afterwards the gender gap was underpinned by elevated mortality among men over the age of 60 [15]. This suggests that the conditions in which women and men live and the risks they are exposed, which are influenced by gender equality, may in turn influence the gender gap in LE.

In keeping with this, previous studies have suggested that gender equality may be associated with the gender gap in LE [16, 17]. Whilst in Europe and the Americas, gender equality appears to be associated with a wider gap in LE, in Africa gender equality may be associated with a narrower gap in LE. These studies used the United Nations Gender Inequality Index [16] and the European Union Gender Equality Index [17] to measure gender equality. However, there is no consensual method of measuring gender equality, and, despite some overlap, each index is based on different domains, which may have different associations with LE. For instance, the United Nations Gender Inequality Index comprises three dimensions: reproductive health (maternal mortality ratio; adolescent birth rate); empowerment (female and male population with at least secondary education; female and male shares of parliamentary seats); and labour market participation (female and male participation rates) [16]. Although it is available for most countries in the world, data are lacking for some countries. Moreover, it provides a relatively simple and, hence, limited assessment of gender equality. Therefore, this study aimed to investigate to what extent gender equality, as measured by the comprehensive index developed by the World Economic Forum, was associated with the gender gap in LE across the globe.

## Methods

### Study design

We conducted an ecological study to investigate the association between gender equality and the gender gap in LE in worldwide. In this ecological study, the unit of analysis was the

population of each country. We adopted a similar methodology to previous studies on this topic [16–18].

## Gender equality

The Global Gender Gap Index (GGGI) was first introduced by the World Economic Forum in 2006 to benchmark progress towards gender parity and compare countries' gender gaps across four dimensions: economic opportunities, education, health, and political leadership. There are three basic concepts underlying the GGGI, forming the basis of how indicators were chosen, how the data are treated and how the scale can be used. First, the index focuses on measuring gaps rather than levels. Second, it captures gaps in outcome variables rather than gaps in input variables. Third, it ranks countries according to gender equality rather than women's empowerment. The GGGI includes four subindexes, each of which is calculated from several indicators (**S1 Table**). The overall GGGI score is a simple average of each subindex score and, similarly to subindex scores, ranges between 1 (parity) and 0 (imparity). Detailed information about the GGGI is available elsewhere [19]. For this study, a modified GGGI (mGGGI) was calculated excluding the health subindex because that included healthy life expectancy, which is the purported outcome of this study. The mGGGI for 2010 to 2020 was used in all the analyses. Data for the years 2006 to 2010 were excluded as estimates were missing for 42 out of 156 countries in 2006 but only for 23 countries in 2010 and this would allow evaluating time trends over a decade [20].

## Life expectancy

Data for LE at birth between 2010 and 2020 were obtained for women and men from The World Bank [21]. Data for LE in 2021 were not available when this study was performed.

## Data analysis

The gender gap in LE was calculated as the difference (values in 2020 minus values in 2010) in LE between women and men for each country. As LE increased for both women and men throughout the study period, a widening or increase in the gender gap in LE meant that LE increased more for women than men, whilst a narrowing or decrease in the gender gap in LE meant that LE increased more for men than women. The mGGGI and its subindices were converted into percentages for ease of interpretation by multiplying the scores by 100. For the purpose of subgroup analysis, countries were grouped into regions as follows: (1) Central Asia and Central and Eastern Europe, (2) North Africa and Middle East, (3) High-income countries, (4) Latin America and the Caribbean, (5) South and Southeast Asia and Oceania, and (6) Sub-Saharan Africa (**S2 Table**). This classification based on geography and development employs the classification used by the Global Burden of Disease [12]. As there is no standard classification of countries into regions, this appeared to be the most appropriate approach because LE and gender equality are likely to be influenced by both socioeconomic development and geographical proximity. Using a purely geographical classification would result in significant heterogeneity within categories, which would render regional measures misleading and bias the association towards the null.

Linear regression was used to estimate the association between change in the mGGGI, and its subindices, and change in the gender gap in LE, LE for women and LE for men between 2010 and 2020. Linear regression was also used to estimate the cross-sectional association between the mGGGI, and its subindices, in 2021 and the gender gap in LE, LE for women and LE for men using the most recent data (from 2020). Simple linear regression models were used with no adjustments. Analyses were performed separately for all subgroups. Gender gap

indices with 95% confidence intervals (CI) were reported and displayed using scatter plots. All analyses were carried out using the base package and all figures constructed using the ggplot2 package for R version 4.1.2. All the code and data needed to replicate this analysis are publicly available on a repository (https://github.com/Ana-Catarina/global-gender-gap.git).

## Results

In total, 156 countries distributed across six world regions were included in this study (**S2 Table**).

### Time trends in the mGGGI and LE

Overall, there was a small improvement in the mGGGI from a mean of 58% in 2010 to 62% in 2021 (**Fig 1**). There were also small improvements from 63% to 66% for the economic subindex and from 16% to 23% for the political subindex. The education subindex remained stable at 95% during the study period. LE increased by about two years for both women and men

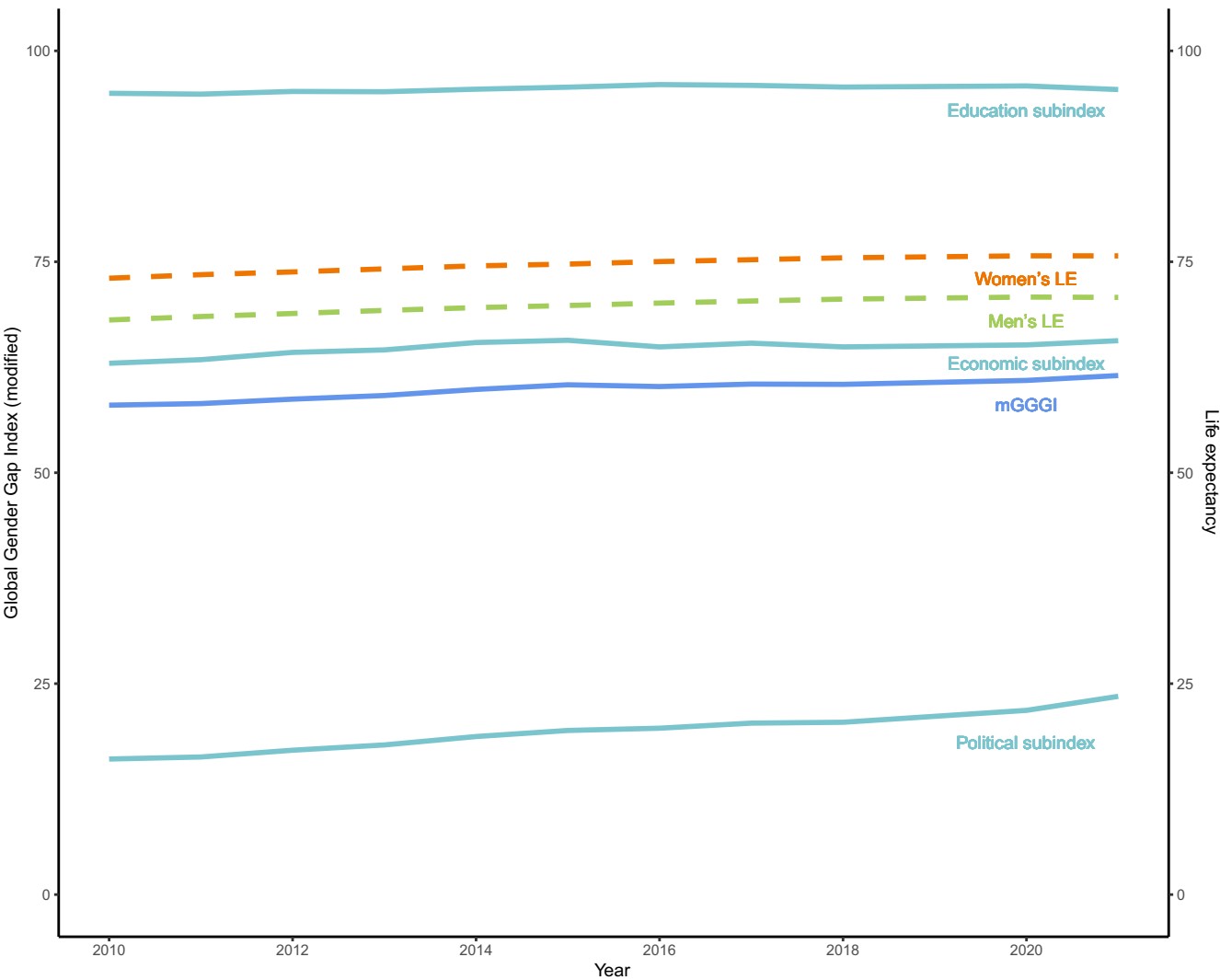

**Fig 1. Time trends in the mGGGI and life expectancy for women and men between 2010 and 2021.**

between 2010 and 2020 (from 75.7 to 77.6 for women and from 69.7 to 71.4 for men), thus maintaining the gender gap in LE relatively constant. In 2021, there was wide variation in the mGGGI between countries from 28% in Afghanistan to 87% in Iceland.

### Association between the mGGGI and LE

**Longitudinal analyses.** Between 2010 and 2020, there was no statistically significant association between the change in the mGGGI and the change in the gender gap in LE (0.00 95% CI [-0.03 to 0.03] years per 1% increase in the mGGGI; p = 0.985) (**Fig 2**). There was also no statistically significant association between the change in the mGGGI and the change in LE for women and men (-0.01 [-0.10 to 0.09] years per 1% increase in the mGGGI, p = 0.913 for women and 0.00 [-0.11 to 0.10] years per 1% increase in the mGGGI, p = 0.928 for men) (**Figs 3 and 4**). The lack of a statistically significant association was comparable across world regions. There was no statistically significant association between the change in the economic and political subindexes and the changes in the gender gap in LE or LE for women and men (**S3 Table**). For the education subindex, each 10% increase between 2010 and 2020 was significantly associated approximately 2.5 and 2-year increases in LE for women and men, respectively, and thus with a 4.5-month widening of gender gap in LE (**S3 Table**).

**Cross-sectional analyses.** In 2021, there was a significant cross-sectional association between the mGGGI and the gender gap in LE, with a 0.7 year (equivalent to 8 months) longer LE per each 10% increase in the mGGGI (0.72 95% CI [0.37 to 1.07] years, p<0.0001; **Fig 5**). There was also a 4.3-month longer LE for women (3.65 [2.48 to 4.82] years, p<0.0001; **Fig 6**)

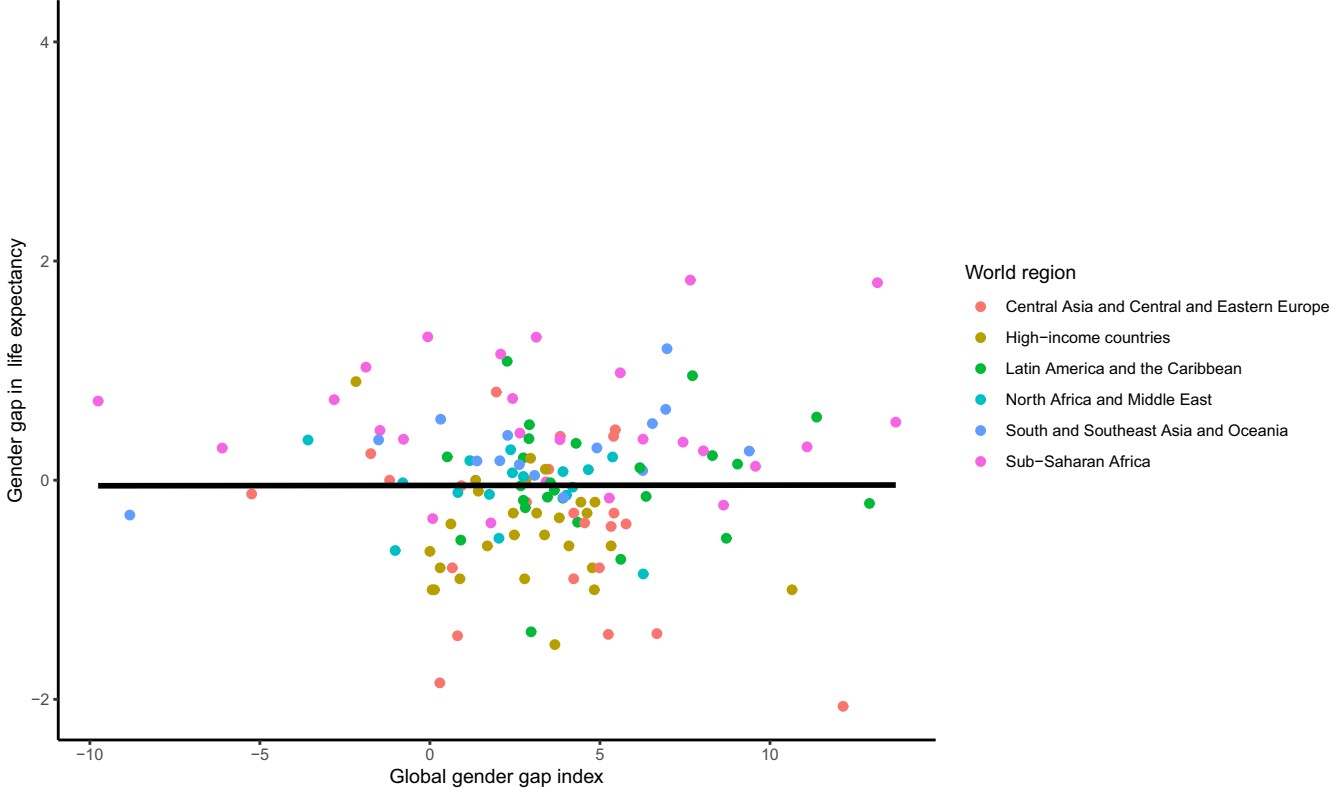

**Fig 2. Association between change in the mGGGI and change in the gender gap in life expectancy between 2010 and 2020.** The scatterplot displays the linear regression line for the association between the change in the mGGGI and the change in gender gap in life expectancy (0.00 [-0.29 to 0.30], p-value 0.985).

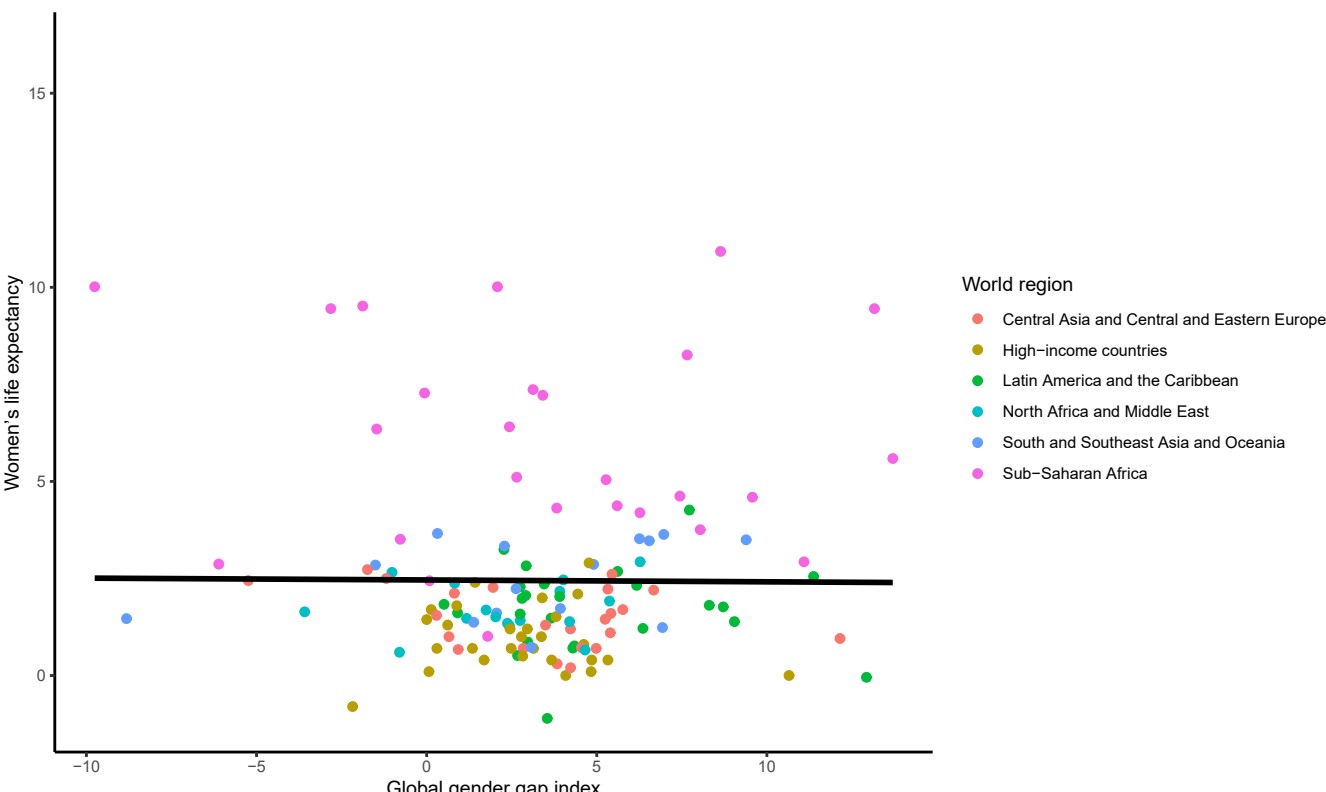

**Fig 3. Association between change in the mGGGI and change in women's life expectancy between 2010 and 2020.** The scatterplot displays the linear regression line for the association between the change in the mGGGI and the change in women's life expectancy (-0.05 [-0.95 to 0.85], p-value 0.913).

and a 3.5-month longer LE for men (2.93 [1.77 to 4.08] years, p<0.0001; **Fig 7**) per each 10% increase in the mGGGI. The economic subindex was not significantly associated with women or men's LE but was associated widening of the gender gap in LE, with each 10% increase in economic equality being associated with a 5.3-month increase in the gender gap in LE (0.44 [0.20 to 0.68] years, p<0.001). The education subindex had a stronger association with women's LE than men's LE (5.04 [3.98 to 6.11] years, p<0.001 for women versus 1.05 [0.69 to 1.41] years, p<0.001 for men per each 10% increase in education equality) (**S4 Table**). Each 10% increase in education equality was associated with a 6-year increase in the gender gap in LE (6.09 [5.08 to 7.11] years, p<0.001). The political index had a similar association with longer LE for women and men (1.56 [0.84 to 2.27] years, p<0.001 for women and 1.45 [0.77 to 2.13] years, p<0.001 for men per each 10% increase in political equality), and hence no statistically significant association with the gender gap in LE.

**Subgroup analysis by world region.** There were significant differences between world regions in the association between the mGGGI and LE in 2020/1 (**S5 Table**). Each 10% increase in the mGGGI was associated with an approximate 6-month reduction in the gender gap in LE in HIC (-0.51 [-0.95 to -0.07] years, p = 0.032). On the other hand, the gender gap in LE increased by about 13 months in South and Southeast Asia and Oceania, and 16 months in Sub-Saharan Africa per each 10% increase in the mGGGI (1.36 [0.63 to 2.09], p = 0.001 and 1.13 [0.29 to 1.96], p = 0.015, respectively). Each 10% increase in the mGGGI was also associated with longer LE for women in North Africa and Middle East (3.55 [1.66 to 5.44], p = 0.002), Sub-Saharan Africa (3.04 [1.01 to 5.08], p = 0.006), South and Southeast Asia and Oceania (2.28 [0.23 to 4.33], p = 0.042).

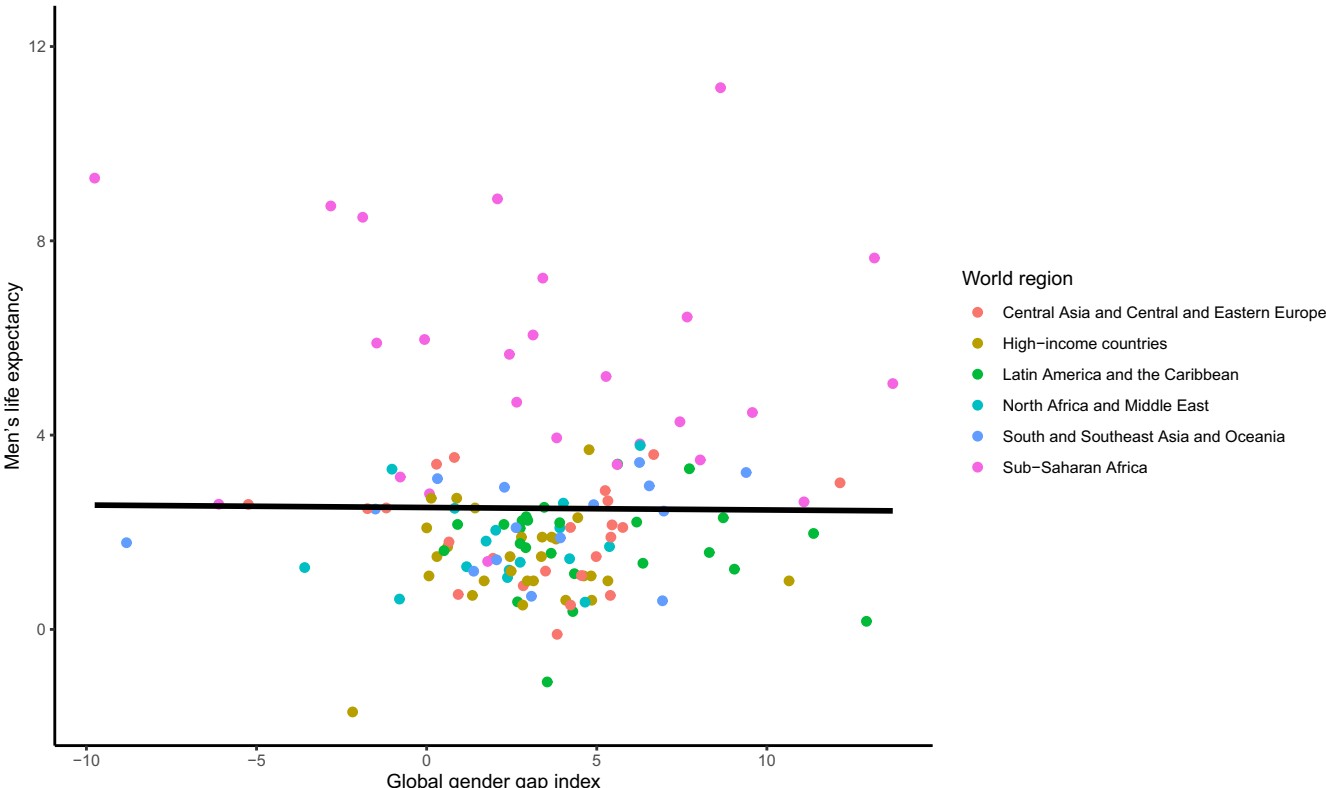

**Fig 4. Association between change in the mGGGI and change in men's life expectancy between 2010 and 2020.** The scatterplot displays the linear regression line for the association between the change in the mGGGI and the change in men's life expectancy (-0.05 [-1.08 to 0.99], p-value 0.928).

There were also differences between regions in the association between the economic and education subindexes of the mGGGI and LE (**S6** and **S7 Tables**) in 2020/1, but the political subindex was not associated with the gender gap in LE or women and men's LE in any region (**S8 Table**). Each 10% increase in economic equality was associated with a narrowing of the gender gap in LE in HIC (about 8 months) and North Africa and Middle East (about 10 months). The gender gap in LE increased by about 6 months in South and Southeast Asia and Oceania and about 18 months in Central Asia and Central and Eastern Europe per each 10% increase in economic equality. The education subindex had the largest association with LE. Each 10% increase in education equality was associated with an approximately 7-year shorter gender gap in LE in HIC. In Central Asia and Central and Eastern Europe and in Sub-Saharan Africa, the gender gap in LE increased by about 7 and 1 year, respectively, per each 10% increment in education equality. Greater education equality was associated with longer LE for women in all regions other than HIC, where each 10% increase in education equality was associated with a 13-year shorter LE for women, and in Latin America and the Caribbean, where there was no association.

## Discussion

This study showed that there was an improvement of 4 percentual points in the mGGGI between 2010 and 2021 from 58 to 62%, which was mainly driven by its political and economic subindexes as the education subindex remained relatively stable. Globally, changes in the mGGGI and its economic and political subindexes were not associated with changes in the

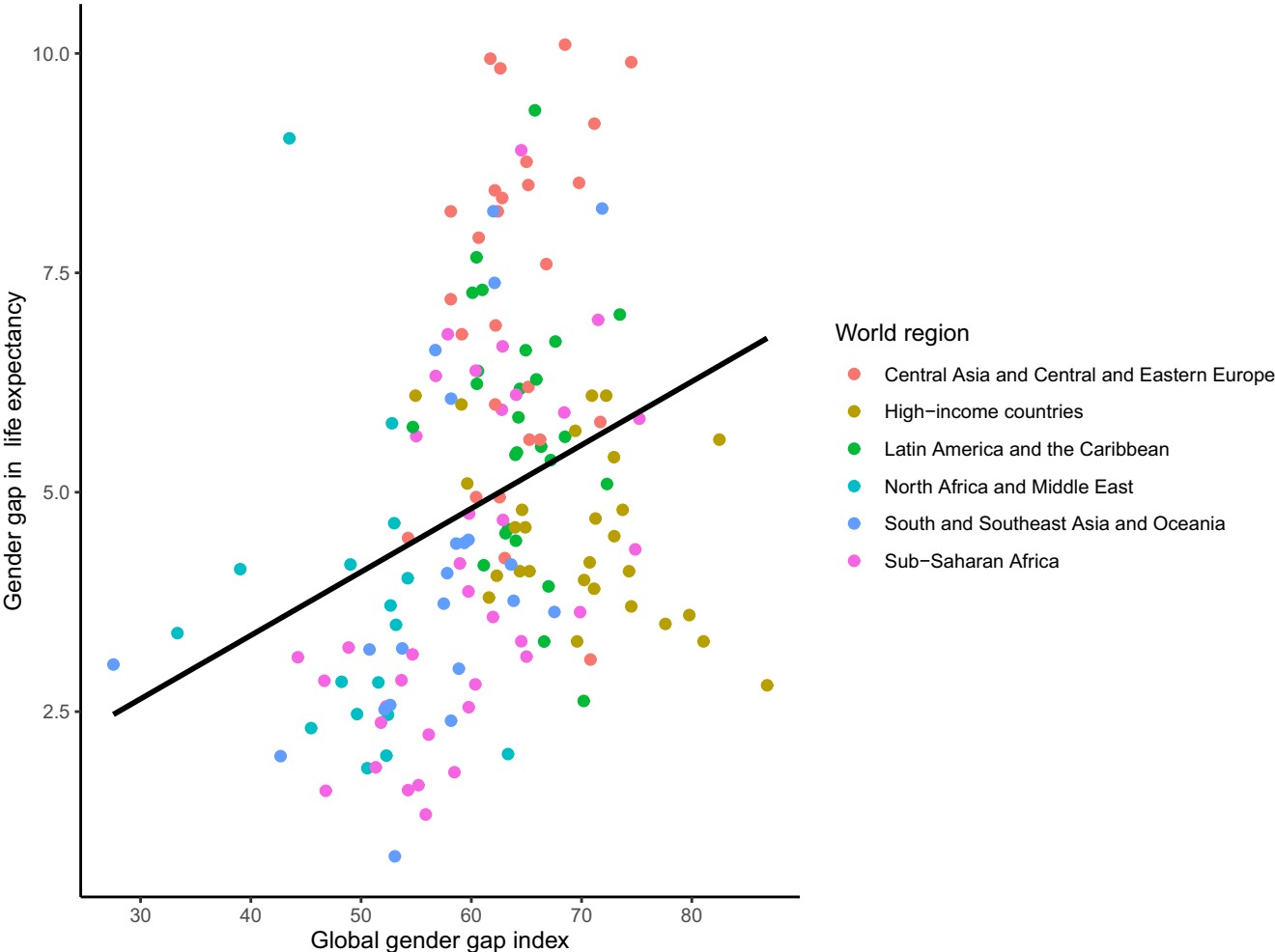

**Fig 5. Cross-sectional association between the mGGGI and gender gap in life expectancy in 2020/1.** The scatterplot displays the linear regression line for the association between the mGGGI and the gender gap in life expectancy (0.72 [0.37 to 1.07], p-value <0.001).

gender gap in LE or with LE for women and men between 2010 and 2020. Only improvements in gender equality in education were associated with a longer LE for women and men and widening of the gender gap in LE, due to a larger increase in LE for women than men. In 2021, each 10% increase in the mGGGI was significantly associated with a 4.3-month increase in women's LE and a 3.5-month increase in men's LE, and hence with an 8-month wider gender gap in LE. However, the direction and magnitude of these associations varied between world regions. Whilst in HIC, the gender gap in LE narrowed by about 6 months per each 10% increase in the mGGGI, in South and Southeast Asia and Oceania as well as in Sub-Saharan Africa the gender gap in LE widened by 13 and 16 months, respectively, per each 10% increase in the mGGGI. Each 10% increase in the mGGGI was also associated with longer LE for women in North Africa and Middle East, Sub-Saharan Africa, South and Southeast Asia and Oceania. In general, the education subindex had the strongest and political subindex had the weakest associations with LE for women and the gender gap in LE.

To the best of our knowledge, this is the first study to investigate the association between the GGGI developed by the World Economic Forum and the gender gap in LE. Our findings

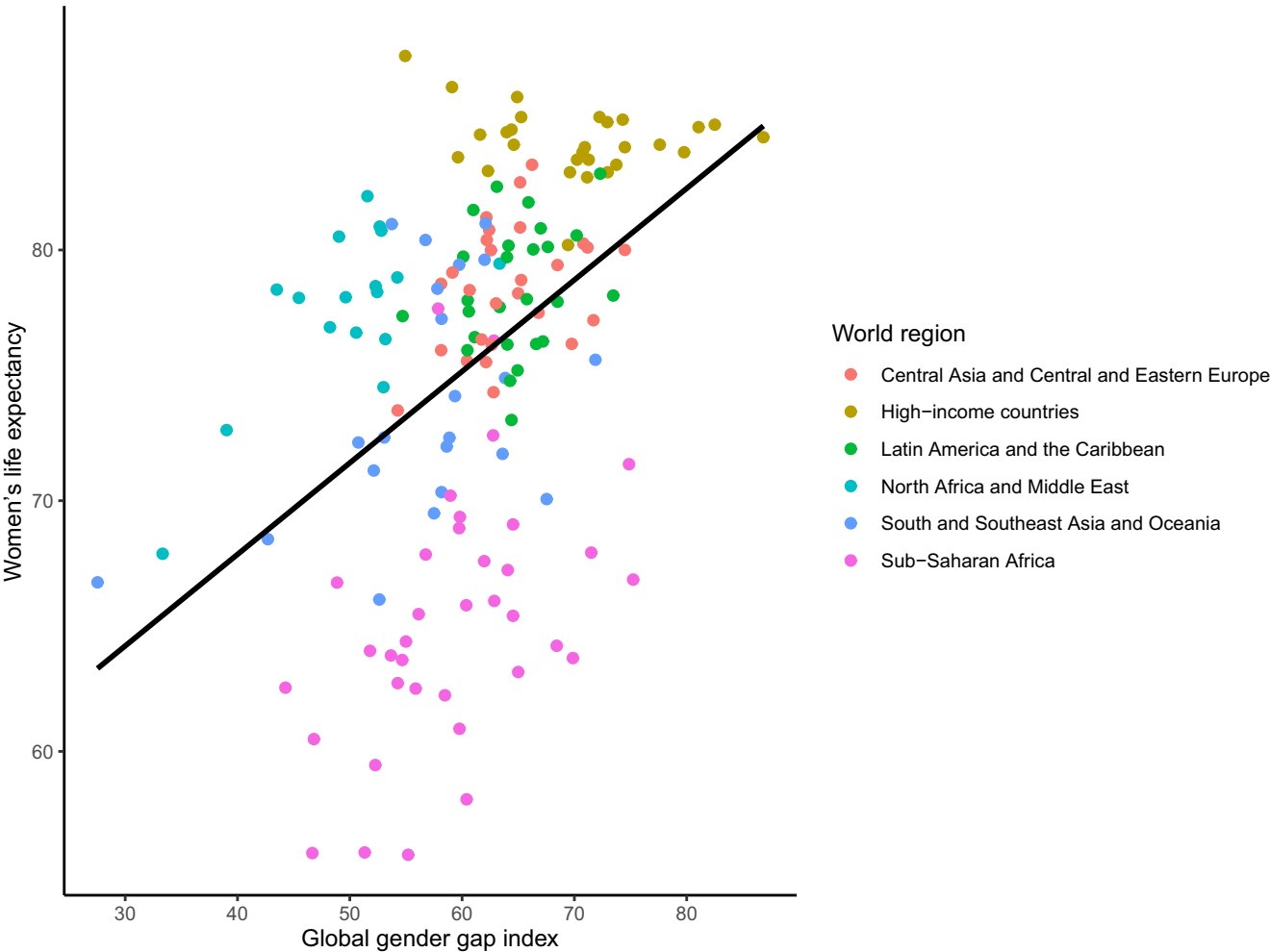

**Fig 6. Cross-sectional association between the mGGGI and women's life expectancy in 2020/1.** The scatterplot displays the linear regression line for the association between the mGGGI and women's life expectancy (3.65 [2.48 to 4.82], p-value <0.001).

are broadly in keeping with a previous study based on the United Nations Gender Inequality Index, which found a positive association between gender inequality and the gender gap in LE for the European region and the Americas, and a negative association in the African region [16]. As we split countries into regions using a classification based on both geography and socioeconomic development, our findings that greater gender equality is overall associated with a narrower gender gap in LE in HIC reflect the fact that this group includes most European and North American countries. In contrast with the previous study, we found an association between greater gender equality and widening of the gender gap in LE not only in Sub-Saharan Africa but also in South and Southeast Asia and Oceania. This discrepancy may be due to the fact that the previous study adopted the United Nations classification of countries into pure geographical regions [16]. In consequence, Asia and Oceania included several HIC and the resulting heterogeneity within these regions would dilute a potential association. Although the previous study adjusted the model for gross national income, democratic status and rural population, we considered those factors were unlikely to have had a material impact in our analyses. First, we divided countries into regions according to socioeconomic

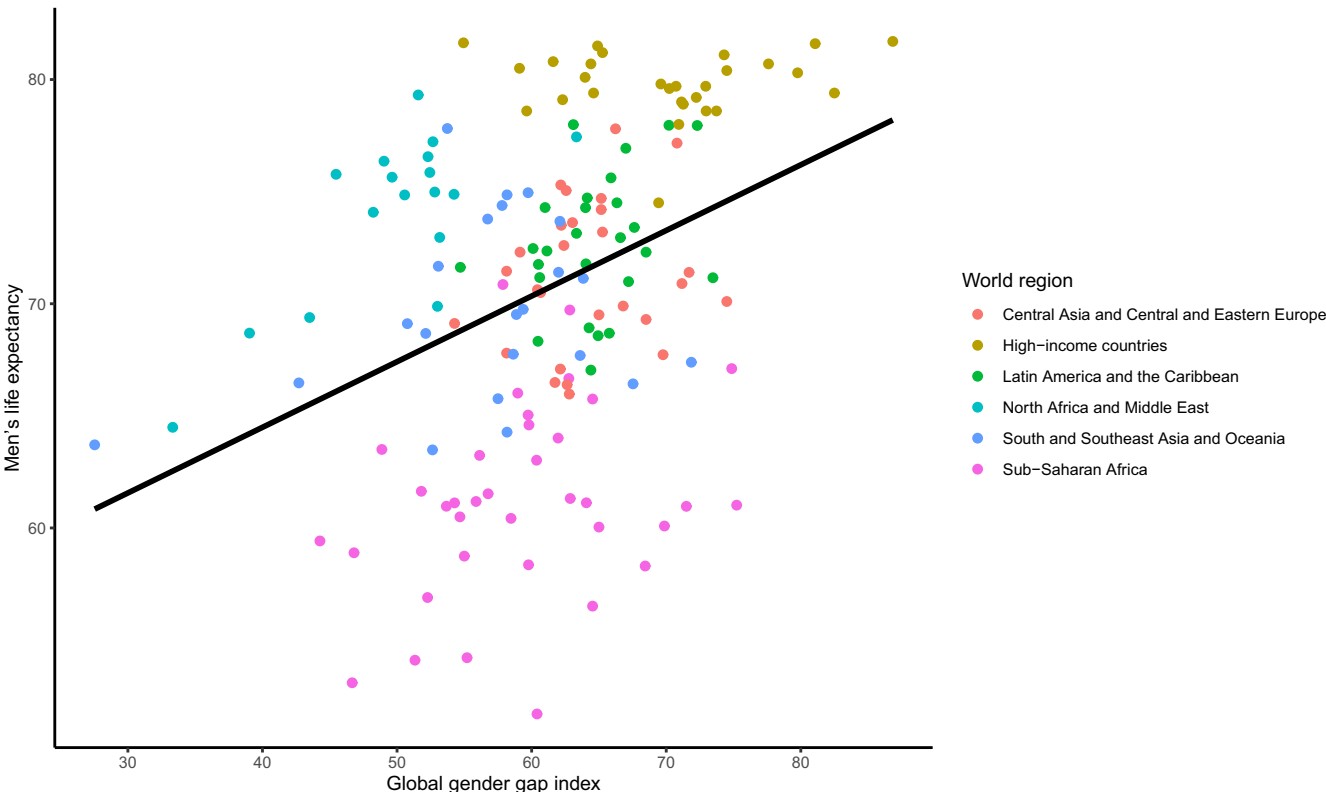

**Fig 7. Cross-sectional association between the mGGGI and men's life expectancy in 2020/1.** The scatterplot displays the linear regression line for the association between the mGGGI and men's life expectancy (2.93 [1.77 to 4.08], p-value<0.001).

development, which allowed us to identify meaningful differences that would otherwise have been overlooked if regions included a heterogeneous mix of countries at different stages of development. Second, we investigated time trends and, hence, those factors could also vary over time. Third, it is not clear why those factors would have a substantially different impact on women versus men, thus resulting in changes in the gender gap in LE.

Our findings of an association between greater gender equality and a narrower gender gap in LE in HIC are in keeping with our previous study in the 27 member states of the European Union, which was based on the Gender Equality Index developed by the European Union [17]. This is perhaps expected as all member states of the European Union were included in the HIC group. The findings of the present study not only corroborate the association using a different index of gender equality but also extend it to several countries outside the European Union, thus suggesting this association is likely to be driven by socioeconomic development rather having a geographical dimension. Although there was a significant association between greater equality in education and shorter LE for women in HIC, this might be due to unknown confounders as the homogeneity in the education subindex among HIC (range 97 to 100%) would make it difficult to find any association.

This study has important implications for policy makers across the globe, particularly as the world gradually recovers from the myriad shocks caused by the COVID-19 pandemic, which had a gendered impact across multiple domains of life [22, 23]. Indeed, the World Economic Forum estimated that closing the global gender gap increased by a generation from 99.5 years to 135.6 years as a result of the COVID-19 pandemic [19]. First, our findings highlight the

importance of gender equality for improving LE, particularly for women in low- and middle-income countries (LMIC). Persisting gender inequalities and lack of empowerment of women in these countries are associated with higher maternal morbidity and mortality, which may be at least partially mediated by lower use of antenatal health services [24, 25]. Second, the discrepancy between HIC and other regions in the association between gender equality and the gender gap in LE suggests that the benefits of gender equality are initially experienced by women, but later extend to men. This may be because women's empowerment and participation in political, economic, and social life has broad benefits at societal level. In fact, evidence demonstrates that enhancing women's representation across multiple sectors contributes to wealthier and, hence, healthier societies for all [26]. Third, our findings that gender equality in education has the strongest association with the gender gap suggests investing in education is paramount. This is particularly important in LMIC, where not only girls are still denied access to education but also resources are limited and, hence, interventions with the highest impact need to be prioritised [27]. Although in 2020 more than two-thirds of countries worldwide had reached gender parity (defined as having a gender parity index value between 0.97 and 1.03) in enrolment in primary education, gender disparities disadvantaging girls in primary education persisted in Africa, the Middle East and South Asia [28]. For instance, 78 girls in Chad and 84 girls in Pakistan were enrolled in primary school for every 100 boys in 2020 [28].

Fourth, the association between gender equality in the economic domain emphasises the importance of addressing the persisting gender pay gap and the low participation of women in the labour market. Across the world, women still get paid 23 per cent less than men and make only 77 cents for every dollar earned by men [29]. In addition, only 49.6% of working-age women are in the workforce in comparison to 76.1% for men [29]. Unleashing the full potential of half the world's population requires gender-sensitive policies and regulations, such as adequate parental leave and flexible hours [30]. However, changing social norms and gendered roles is critical to achieve economic equality. Even in HIC, women contribute more to household chores and childcare and less to the workplace than men, and this has a detrimental impact on their health [31, 32]. Despite the benefits of financial independence afforded by employment, evidence from LMIC shows that men remain household authority figures, including for health decision making, thus highlighting the need to address cultural and social norms that hinder gender equality [33]. Fifth, the lack of association between gender equality in the political subindex and the gender gap in LE raises concerns about the extent to which tokenistic approaches to gender equality are being implemented by political systems worldwide. Gender quotas, although instrumental to change societal values and encourage women's participation in politics, have not yet achieved gender parity in political parties and parliaments, even in HIC [34]. Furthermore, even in political organisations where women are present in large numbers, glass ceilings often remain firmly in place, thus limiting women's power and ability to influence policy. Women still experience significant challenges as politicians, such as addressing discrimination or cultural beliefs that limit women's role in society; balancing private, family and political life; gaining support from political parties; and securing campaign funding [35]. They may also face violence, harassment, and intimidation or be dissuaded from running for office [35]. Ultimately, this means that women still lack the power to influence policy and decision making even when numerically represented in political systems, which might explain we found no association between gender equality in politics and LE.

The main strengths of this study are the fact it included a large number of countries and relied on reliable data provided by the World Economic Forum. This allowed demonstrating how associations between LE and gender equality differ across different world regions. By drawing on an index that combined different aspects of gender equality, this study was able to

investigate which factors have the strongest association with LE, which can help setting priorities in public health policy. In addition, splitting countries according to socioeconomic development and geography enabled identifying differences between regions that are important for policy makers across the globe. However, there are also some limitations. First, data were not available for all the countries in the world, which means the associations observed in some regions might be different if all the countries in that region were included. Second, it is possible the observed associations between gender equality and LE are explained by other factors, such as socioeconomic development, sociocultural norms, or other unmeasured confounders. Third, although the mGGGI includes important drivers of gender inequalities, it is not exhaustive and inclusion of additional indicators of gender inequalities could have changed the study findings.

## Conclusions

Globally, greater gender equality is associated with longer LE for both women and men. Greater gender equality is also associated with a widening of the gender gap in LE, due to a larger increase in LE for women than men. The variation in this association across world regions defined on socioeconomic development and geographic proximity suggests that gender equality may initially widen the gender gap in LE as the benefits of greater gender equality benefit mainly women's lives and health. As countries progress along the continuum of gender equality, the benefits of increased participation of women in society extend to men, thus leading to a larger increase in men's LE and a narrowing of the gender gap in LE. Therefore, addressing longstanding gender inequality and empowering women might help extending longevity for both women and men.

## Supporting information

**S1 Table. Subindexes included in the global gender gap index.**
(DOCX)

**S2 Table. Categorisation of countries into regions.**
(DOCX)

**S3 Table. Association between change in the mGGGI and its subindexes and change in LE for women and men and gender gap in LE between 2010 and 2020.**
(DOCX)

**S4 Table. Cross-sectional association between the mGGGI and its subindexes and LE for women and men and gender gap in LE in 2021.**
(DOCX)

**S5 Table. Cross-sectional association between the mGGGI and LE for women and men and gender gap in LE stratified by region in 2021.**
(DOCX)

**S6 Table. Cross-sectional association between the economic subindex of the mGGGI and LE for women and men and gender gap in LE stratified by region in 2021.**
(DOCX)

**S7 Table. Cross-sectional association between the education subindex of the mGGGI and LE for women and men and gender gap in LE stratified by region in 2021.**
(DOCX)

**S8 Table. Cross-sectional association between the political subindex of the mGGGI and LE for women and men and gender gap in LE stratified by region in 2021.**
(DOCX)

## Author Contributions

**Conceptualization:** Sanne A. E. Peters, Mark Woodward.

**Data curation:** Ana-Catarina Pinho-Gomes.

**Formal analysis:** Ana-Catarina Pinho-Gomes.

**Investigation:** Ana-Catarina Pinho-Gomes.

**Methodology:** Ana-Catarina Pinho-Gomes, Sanne A. E. Peters, Mark Woodward.

**Resources:** Ana-Catarina Pinho-Gomes.

**Supervision:** Sanne A. E. Peters, Mark Woodward.

**Visualization:** Ana-Catarina Pinho-Gomes.

**Writing – original draft:** Ana-Catarina Pinho-Gomes.

**Writing – review & editing:** Ana-Catarina Pinho-Gomes, Sanne A. E. Peters, Mark Woodward.

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
