## [Decision Letter · Decision Letter 0]

27 Oct 2022

PGPH-D-22-01210

Gender equality related to gender differences in life expectancy across the globe Gender equality and life expectancy

Dear Dr Ana-Catarina Pinho-Gomes

Thank you for submitting your manuscript to PLOS Global Public Health. After careful consideration, we feel that it has merit but does not fully meet PLOS Global Public Health’s publication criteria as it currently stands. Therefore, we invite you to submit a revised version of the manuscript that addresses the points raised during the review process.

We look forward to receiving your revised manuscript.

Kind regards,

Seth Christopher Yaw Appiah, PhD PhD

Academic Editor

Journal Requirements:

2. Please send a completed 'Competing Interests' statement, including any COIs declared by your co-authors. If you have no competing interests to declare, please state "The authors have declared that no competing interests exist". Otherwise please declare all competing interests beginning with the statement "I have read the journal's policy and the authors of this manuscript have the following competing interests:"

3. Your current Financial Disclosure states, “This study was not funded. MW is supported by National Health and Medical Research Council of Australia grants APP1149987 and APP1174120. CC is funded by the National Heart Foundation of Australia (Postdoctoral fellowship 102741). ACPG is funded by an Academic Clinical Fellowship by the National Institute for Health Research. SAEP is supported by a UK Medical Research Council Skills Development Fellowship (MR/P014550/1)”. However, your funding information on the submission form indicates that you did not received any funding. Please indicate by return email the full and correct funding information for your study and confirm the order in which funding contributions should appear. Please be sure to indicate whether the funders played any role in the study design, data collection and analysis, decision to publish, or preparation of the manuscript.

4. Please provide separate figure files in .tif or .eps format only and remove any figures embedded in your manuscript file. Please also ensure that all files are under our size limit of 10MB.

5. We have noticed that you have uploaded Supporting Information files, but you have not included a list of legends. Please add a full list of legends for your Supporting Information files after the references list. 

6. In the online submission form, you indicated that "Data are available upon request from the corresponding author". All PLOS journals now require all data underlying the findings described in their manuscript to be freely available to other researchers, either 1. In a public repository, 2. Within the manuscript itself, or 3. Uploaded as supplementary information.

7. Fig 2: please (a) provide a direct link to the base layer of the map (i.e., the country or region border shape) and ensure this is also included in the figure legend; and (b) provide a link to the terms of use / license information for the base layer image or shapefile. We cannot publish proprietary or copyrighted maps (e.g. Google Maps, Mapquest) and the terms of use for your map base layer must be compatible with our CC-BY 4.0 license. 

Reviewers' comments:

Reviewer's Responses to Questions

**Comments to the Author**

1. Does this manuscript meet PLOS Global Public Health’s publication criteria? Is the manuscript technically sound, and do the data support the conclusions? The manuscript must describe methodologically and ethically rigorous research with conclusions that are appropriately drawn based on the data presented.

Reviewer #1: Partly

Reviewer #2: Yes

2. Has the statistical analysis been performed appropriately and rigorously?

Reviewer #1: No

Reviewer #2: I don't know

3. Have the authors made all data underlying the findings in their manuscript fully available (please refer to the Data Availability Statement at the start of the manuscript PDF file)?

Reviewer #1: No

Reviewer #2: Yes

4. Is the manuscript presented in an intelligible fashion and written in standard English?

Reviewer #1: Yes

Reviewer #2: Yes

5. Review Comments to the Author

Reviewer #1: - This is an important and outstanding topic of research. However, the novelty of the study was not highlighted! Adding the novelty of your research to the introduction section would underline its importance.

- Data analysis on R is not available, making the results presented by the authors questionable, especially since the study shows outstanding results. For example: "there was no association between the change in the mGGGI and the change in the gender gap in LE."

- I would recommend discussing the definition of the "gender equality" term and how it might be related to LE according to the literature.

- The literature review is missing, which is essential to understand the reason behind the adapted methods and methodology in this study. How would we know that the applied methodology is appropriate for this type of study without a relevant literature review?

- More discussion is required for the Study Design section. What is an ecological study? It seems more like a description of the study than about methodology

- More discussion is required on the four sub-indices and how changes in these sub-indices are reflected on GGGI. The authors mentioned in the Discussion section, "which is mainly driven by its political and economic subindexes as the education subindex remained relatively stable", which means that the discussion is related to how these sub-indices change the GGGI.

- Ref 10: could you please refer to the page number in the attributed reference? I tried to explore more information about the GGGI in the reference, but it was an uneasy task to refer directly to the target section in the reference, which would be the same for the readers.

- Gender equality section: "The MGGG for 2010 to 2020 .... evaluating time trends over a decade." there is no reference.

- LE data are missing (for women, men, and the gender gap in LE calculation).

- Life Expectancy section: "Data for LE at birth between .... from the World Bank. Data for LE in 2021 were not available" there is no reference.

- It seems like there is missing information before the Data Analysis section. The authors mentioned that the mGGGI and its sub-indices were converted into percentages. What are the data that they were converted? Where can the reader find them?

- I would recommend moving lines 103 to 110 to the Study Design section, where more discussion about the research methods and variables is required.

- Data of the mGGGI and its sub-indices after converting into percentages and standardizing them are missing.

- The categorization of the countries into regions is not referenced.

- Why do the authors specifically use the Global Burden of Diseases categorization of countries?

- The method the results were presented is inconsistent and confusing. I would suggest rephrasing this section.

- In the Discussion section, the authors justified the discrepancy by the fact that there are several HIC in the region of Asia and Oceania. However, referring back to the categorization of the countries in the Data Analysis section, I would presume that HIC is a separate group in terms of data analysis.

- Lines 231 to 278 sound irrelevant to the Discussion section. The authors presenting subjective opinions about the importance of the study. This paragraph seems to be more about the introduction of the study after major rephrasing.

- The reason behind categorizing the countries into multiple groups is mentioned in the Discussion section. I would recommend moving it to the method section.

Reviewer #2: The research study investigated the association of gender equality with the gender gap in life expectancy (LE) across the globe. The study uses a modified version of the Global Gender Gap Index - excluding the health subindex because it included life expectancy. The longitudinal analyses did not find an association between the change in the mGGGI and the change in the gender gap in LE. There was also no association between the change in the mGGGI and the change in LE for women and men. While there was no association between the change in the economic and political subindexes and changes in the LE gender gap or LE for women and men, there was a widening of the LE gender gap for the education subindex. There were some more associations found in the cross-sectional analyses. These were explained well in the discussion section. It would be helpful if authors clarified what an "increase in mGGGI" means.

This research study has great potential for publication. There are some minor edits for consideration in order to strengthen and tighten the reporting of the study. One, it would be helpful to clarify in which direction the gender gap is widening, throughout the manuscript. It is my understanding that there are several ways the gender gap can increase: increase in LE for women while it remains the same for men; decrease in LE for men, while staying the same for women; an increase in LE for men and women, with a greater increase for women; or an increase for women and a decrease for men. It would be helpful to clarify the reason for each difference in the gap whenever it is mentioned throughout the manuscript, or clarify terminology so that the reader can understand as they are reading.

The authors may want to consider linking the ideas in the introduction more with each other in order to develop the introduction further. Additionally, the authors may want to include references in the introduction that explain life expectancy and its role as a measure in public health from larger institutions (WHO, WEF, etc.) in addition to the citations already included.

The authors may want to write that there is no statistically significant association when writing in the results section, rather than that there is no association, wherever applicable. I'm not sure whether this study could be replicated based on the information that is provided in the methods section. The authors may want to consider providing more details about the setup of the linear regression models.

The discussion section is well-developed and explains the results well. The parts of the discussion with the policy recommendations would be made stronger if it included more support from literature. When writing recommendations such as "investing in education is paramount", "this is particularly important in LMIC", and suggestions for parental leave and flexible hours, the statements would be made even stronger if they had support from existing literature. Each recommendation and statement that includes information outside of the study should be supported by literature.

In the discussion section, the "lack of association between gender equality in the political subindex and the gender gap in LE" and the implications for policy was a little confusing; I would suggest making that connection more clear. Given that the mGGGI includes economic opportunity, I'm not sure why one of the included limitations is that the observed associations between gender equality and HLE and LE are explained by other economic factors, even if the analysis didn't suggest it. This may be a major limitation.

The conclusion section makes statements that are not mentioned in the literature (i.e. ideal measures and markers for gender equality and rate of progress in the past decade) so should focus more on the findings and literature that are mentioned in the manuscript, or include this evidence in the introduction or the discussion. It would also be best to omit any informal language such as "long way off" and use technical language (that is still accessible for people to understand).

The figures should show the magnitude of the association within the figure.

In general, there were a few typos throughout the manuscript, so the authors should check for grammar, typos, and clarity throughout the manuscript.

Overall, this is a very important topic and I am glad that you all undertook this study. Thank you for sharing this research!

6. PLOS authors have the option to publish the peer review history of their article (what does this mean?). If published, this will include your full peer review and any attached files.

**Do you want your identity to be public for this peer review?** For information about this choice, including consent withdrawal, please see our Privacy Policy.

Reviewer #1: No

Reviewer #2: No

---

## [Editor Report · Decision Letter 1]

17 Jan 2023

Gender equality related to gender differences in life expectancy across the globe Gender equality and life expectancy

PGPH-D-22-01210R1

Dear Dr Ana-Catarina Pinho-Gomes

We are pleased to inform you that your manuscript 'Gender equality related to gender differences in life expectancy across the globe Gender equality and life expectancy' has been provisionally accepted for publication in PLOS Global Public Health.

Best regards,

Seth Christopher Yaw Appiah, PhD,PhD

Academic Editor